# Adipokines as Biomarkers of Atopic Dermatitis in Adults

**DOI:** 10.3390/jcm9092858

**Published:** 2020-09-04

**Authors:** Andrzej Kazimierz Jaworek, Jacek C. Szepietowski, Krystyna Szafraniec, Magdalena Jaworek, Przemysław Hałubiec, Anna Wojas-Pelc, Mieczysław Pokorski

**Affiliations:** 1Department of Dermatology, Jagiellonian University Medical College, 31-008 Cracow, Poland; anna.wojas-pelc@uj.edu.pl; 2Department of Dermatology, Venereology and Allergology, Wroclaw Medical University, 50-367 Wroclaw, Poland; jacek.szepietowski@umed.wroc.pl; 3Department of Epidemiology and Population Studies, Institute of Public Health, Faculty of Health Sciences, Jagiellonian University Medical College, 31-008 Cracow, Poland; mygomola@cyf-kr.edu.pl; 4Department of Physiotherapy, Faculty of Health Sciences, Jagiellonian University Medical College, 31-008 Cracow, Poland; magdalena.1.jaworek@uj.edu.pl; 5Student Scientific Group, Department of Dermatology, Jagiellonian University Medical College, 31-008 Cracow, Poland; przemyslawhalubiec@gmail.com; 6Faculty of Health Sciences, University of Opole, 45-040 Opole, Poland; m_pokorski@hotmail.com

**Keywords:** adipokines, adiponectin, atopic dermatitis, biomarkers, eczema, pruritis

## Abstract

Atopic dermatitis (AD) is characterized by chronic, relapsing, pruritic skin inflammation and does not have a well-understood pathogenesis. In this study, we addressed the contribution of adipokines to AD eczema based on the assessment of blood levels of adiponectin, resistin, leptin, lipocalin-2, and vaspin in adult non-obese patients suffering from chronic extrinsic childhood-onset AD. We investigated 49 AD patients with a median age of 37 years. The control group consisted of 30 age-matched healthy subjects. Adipokines were assessed in the serum by ELISA assays and the severity of AD with the SCORing Atopic Dermatitis (SCORAD) index. We found that adiponectin and resistin decreased and leptin appreciably increased in AD patients when compared to those in healthy subjects. Further, the levels of adiponectin and resistin were inversely related to the intensity of eczema. In conclusion, apart from the formerly investigated role of leptin in AD, this study points to adiponectin and resistin as the potential candidate adipokine biomarkers involved in shaping eczema intensity and severity, which may help predict disease exacerbations and enable the development of effective targeted therapeutic interventions.

## 1. Introduction

Atopic dermatitis (AD) is a chronic disease that may occur at any age. The universal symptom is skin inflammation associated with intense pruritis. Eczema flares periodically, which degrades quality of life and is a severe socioeconomic detriment [1]. AD is now the most prevalent inflammatory disease worldwide. It also increases the risk of other skin diseases, particularly infections due to an impaired skin barrier or other atopic disorders such as asthma, food allergies, and the like. There are also non-atopic AD comorbidities such as contact dermatitis, autoimmunologic disorders, and psychiatric disorders, particularly depression, as well as cardiovascular disorders [2,3,4]. The association of AD with other disorders is considered bidirectional but the underlying pathophysiological mechanisms are unclear. A causal relationship is often put into question in case of brain and heart infarcts or coronary heart disease, which may stem from disordered sleep architecture and obesity, which are frequent accompaniments of AD [5]. The disease increasingly concerns adults, rather than being a disease of childhood only. A recent American National Health Interview Survey, conducted with 35,000 adults, has shown a 7.2% prevalence of eczema/skin allergy [6]. Likewise, in the European Community Respiratory Health Survey, AD prevalence reached 6.2% in a population of 8206 adults [7]. The results of the Epidemiology of Allergic Diseases study in Poland show that the prevalence of AD exceeds 3.6% in adults [8]. An extensive international study by Barabarot et al. [9] in population samples from the US, Japan, Canada, and Europe shows the prevalence of past or current active AD ranging from 2.1% to 4.9%. Raciborski et al. [10] have demonstrated that AD is a cause of nearly 260,000 specialist consultations and 8000 hospitalizations in Poland annually.

The pathogenesis of eczema remains enigmatic, which hinders treatment modalities [11,12]. A common and long-standing presumption of the essential role of histamine and protease release from mast cells during their allergen-induced immunoglobulin E (IgE)-mediated degradation in evoking skin itch has not stood the test of time in the face of an insufficient therapeutic effect of anti-histamines [13]. On the other hand, neuroactive peptides (exemplified by substance P), vasoactive intestinal peptide, or neurotensin, which provoke neurogenic skin inflammation, eczema, and itch, are at play in AD [14]. A spate of cytokines such as interleukins (IL), tumor necrosis factor-alpha (TNF-α), interferon-gamma (INF-γ), and others are also released from T-lymphocytes and other components of the adaptive immune system, although their pathogenetic role in the itch of AD is unclear or contentious. Nonetheless, some of them, like IL-2, have an identifiable itch-enhancing effect [15]. Highly effective biological therapy with dupilumab, a monoclonal antibody that binds to IL-4 and IL-13 receptors, substantiates the role of interleukins in the development of AD symptoms [16].

Dysfunction of the skin barrier, both in lesional and non-lesional skin, plays a fundamental role in the pathogenesis of AD. Keratinocytes in the barrier-disrupted skin accelerate the type 2 immune response by producing alarmins, like thymic stromal lymphopoietin, IL-25, and IL-33 [11,17]. The thymus activation-regulated chemokine is a frequently considered biomarker of AD. However, its usefulness for predicting disease aggravation is limited due to low sensitivity and specificity [18]. In the search for more suitable biomarkers of AD, attention has been turned to a family of signaling proteins called adipokines, which are secreted by visceral and subcutaneous adipose tissue. These proteins, aside from the traditional environmental and lifestyle metabolic components related to the lack of physical activity and overnutrition in obesity and diabetes, are upregulated in immunity-related disorders (i.e., psoriasis) that frequently accompany obesity [19,20]. This group of cytokines includes leptin, vaspin, lipocalin-2, and resistin, which are mostly proinflammatory. Leptin, which is involved in satiety and energy processes, may be considered the archetype of these cytokines. It stimulates the production of IL-6 and TNF-α from adipose tissue and promotes CD4+ T lymphocyte activation toward the Th1 phenotype [19,20]. Leptin also inhibits the infiltration of the adipose tissue by regulatory T lymphocytes (Tregs) that increase immune tolerance, and by so doing it has a pro-inflammatory action at the tissue level [21]. Serum leptin has been reported to be increased [22] or grossly unchanged in childhood AD [23], but there is a lack of confirmatory studies in adult AD. Adiponectin, on the other hand, plays an anti-inflammatory role in immune and inflammatory disorders and it inversely correlates with psoriasis severity [19]. Adiponectin appears to decrease in AD [24], but this finding was not conclusively confirmed in another study [25].

Aside from psoriasis, obesity increases the propensity for AD [20], but the potential role of adipokines in AD is unclear, contentious, and seldom considered or investigated. Even less is known about the role of adipokines in the development of AD in non-obese individuals. Therefore, in this study, we address the role of adipokines in AD by assessing the serum levels of a set of cytokines outlined above in adult non-obese patients suffering from chronic eczema.

## 2. Patients and Methods

The study was approved by the Bioethics Committee of the Jagiellonian University in Cracow, Poland (permit: 122.6120.117.2016). All individual participants gave written informed consent to participate in the study. We investigated 49 adult Caucasian patients suffering from childhood-onset AD (male/female (M/F) 17/32; median age: 37; range: 20–75 years) who were hospitalized at the Dermatology Department of Jagiellonian University Medical College in 2017. The control group consisted of 30 healthy sex and age-matched subjects (M/F): 15/15; median age: 42; range: 22–78 years). In both groups, exclusion criteria were as follows: age below 18 years, pregnancy or lactation, current additional allergic symptoms (such as asthma and allergic rhinitis) other than cutaneous AD or systemic inflammatory disorders, cardiovascular and metabolic disorders, immunosuppressive treatment no later than three months before the study, and phototherapy no later than six months before the study.

The diagnosis of AD was established based on the Hanifin and Rajka criteria [26]. Extrinsic AD was defined as fulfilling one of two criteria: total IgE > 100 IU/mL, or the presence of specific serum IgE (or skin prick test positive) for specific air- or food-derived allergens [1,4]. A detailed anamnesis concerning the history of AD was taken and each patient was subjected to clinical examination. The severity of AD was based on the SCORing Atopic Dermatitis (SCORAD) index, where a score under 25 is defined as mild eczema and over 50 points as severe eczema [27]. The maximum intensity of itching during the preceding 24 h was evaluated on a visual analog scale (VAS), where 0 defined no itch, 1–3 mild itch, 4–6 moderate itch, 7–8 severe itch, and 9–10 very severe itch [28]. The percentage of patients suffering from varying degrees of itch severity was also assessed [29]. The basic characteristics of the patients are presented in Table 1.

Blood samples (10 mL) were drawn from the ulnar vein after a 12-h night-time fast and a 60-min rest between 7:00 am and 8:00 am. The samples were kept for 2 h at room temperature for clotting, then centrifuged at 3500 rpm for 10 min and frozen at −80 °C until use. Serum levels of the adipokines leptin, lipocalin-2, resistin, adiponectin, and vaspin were determined using commercially available kits according to the manufacturer’s instructions via a standard enzyme-linked immunosorbent assay (ELISA, R&D System, Minneapolis, MN, USA). The assay’s detection levels were as follows: adiponectin: 0.891 ng/mL; resistin: 0.055 ng/mL; lipocalin-2: 0.040 ng/mL; leptin: 0.128 ng/mL; and vaspin: 14.600 pg/mL. All measurements were performed in duplicate for each patient and the average of the two was taken as the final result. Total IgE (tIgE) concentration was routinely assessed using the ELISA method and a UniCAP fluorimeter (Pharmacia, Stockholm, Sweden). The upper limit of normal IgE was taken as 100 IU/mL.

### Statistical Elaboration 

Data were expressed as medians and minimum–maximum ranges. Since the distribution of cytokine results was skewed, the non-parametric Mann–Whitney test was used to assess the significance of differences in the serum content of adipokines between AD patients and the control group. Non-parametric ANOVA was used for differences among the groups of mild and severe eczema, and controls. Spearman’s rank correlation was used to determine the strength and direction of a relationship between adipokines. Linear regression was used to test for trends concerning the effect of each category of eczema severity on the outcome, i.e., the median values of adiponectin, compared with the reference level (P_trend_) [30]. The level of statistical significance was α = 0.05. Statistical analysis was performed using a commercial statistical package, SPSS v24.0 (IBM Corporation, Armonk, NY, USA).

## 3. Results

All patients were classified as having extrinsic AD. The median age of patients with severe AD was greater than that of those with mild AD (41 vs. 28 years, respectively; *p* < 0.05). Most patients (30/49; 61%) suffered from severe AD, with a median VAS pruritis score of 9 (min–max: 8–10 points). Only three of these patients were overweight. The mean BMI, however, was not different between the groups of patients with mild and severe AD (*p* >0.05). A BMI over 25 kg/m^2^ was not observed in the patients with SCORAD < 25 points (mild AD). The median tIgE in the whole group of patients was 2300 IU/mL (min–max: 105–58,300 IU/mL) (Table 1).

### 3.1. Adiponectin

Compared to healthy controls, AD patients showed a significantly lower content of serum adiponectin (*p* = 0.020) (Table 2). Although there were no differences in the adiponectin content between mild and severe eczema, a decreasing trend was noticed with increasing severity of eczema (P_trend_ = 0.005) (Figure 1). Adiponectin content did not correlate with sex, age, BMI, tIgE level, or other adipokines (Table 3).

### 3.2. Resistin

Serum resistin was significantly lower in AD patients compared to healthy controls (*p* < 0.001) (Table 2; Figure 2) and it progressively decreased with eczema severity as determined by SCORAD (r = −0.38; *p* = 0.008). There were insignificant differences in serum resistin between mild eczema (median: 11.8 ng/mL; min–max: 9.7–18.4 ng/mL) and severe eczema patients (median: 10.8 ng/mL; min–max: 7.4–16.8 ng/mL) (*p* = 0.090). Serum resistin was associated with lipocalin-2 changes when considering the whole AD group (r = 0.37; *p* = 0.009) (Table 3); the association was notably strengthened in patients with severe eczema (r = 0.60; *p* = 0.003). An inverse correlation was also noticed between resistin and tIgE (r = −0.36; *p* = 0.012).

### 3.3. Lipocalin-2

There were insignificant differences in serum lipocalin between mild eczema (median: 83.0 ng/mL; min–max: 55.6–147.2 ng/mL), severe eczema patients (median: 87.8 ng/mL; min–max: 56.0–147.6 ng/mL), and healthy controls (median: 96.7 ng/mL; min–max: 52.6–178.3 ng/mL) (*p* = 0.714), other than the afore-mentioned association between lipocalin-2 and resistin levels. 

### 3.4. Leptin

Serum leptin was significantly higher in the whole AD group compared to that in healthy controls (*p* = 0.02) (Table 2), with no differences depending on eczema severity (mild eczema median: 10,701.7 ng/mL, min–max: 1914.1–35,158.3 ng/mL; severe eczema median: 8392.7 ng/mL, min–max: 2000.1–76,299.8 ng/mL) (*p* = 0.064). Leptin content failed to associate with sex, age, BMI, tIgE, or with the other adipokines.

### 3.5. Vaspin

There were insignificant differences in serum vaspin between AD patients, irrespective of eczema severity, and controls (*p* = 0.540) (Table 2). No differences in serum vaspin were noticed depending on eczema severity (mild eczema median 238.4 pg/mL, min–max: 113.2–877.6 pg/mL; severe eczema median 210.7 pg/mL, min–max: 93.5–891.7 pg/mL (*p* = 0.739)), nor were there any associations between vaspin and sex, age, BMI, total IgE, or with other adipokines.

## 4. Discussion

This study was designed to examine the level of several circulating adipokines that could be involved in the complex neuroendocrine–immune–metabolic networks of signal communication underlying metabolic skin homeostasis. Disordered skin homeostasis is the foundation of AD eczema, a disease whose incidence in adults is on the rise worldwide and does not seem related to obesity as originally thought. The major finding of this study was that serum adiponectin and resistin decreased and leptin appreciably increased in adult patients suffering from extrinsic AD when compared to control healthy subjects. Further, the levels of adiponectin and resistin were inversely related to the intensity of eczema assessed using the SCORAD index. Thus, apart from the previously studied role of leptin in AD, this study directs attention to adiponectin and resistin as the candidate adipokines possibly involved in shaping eczema severity, which may help predict disease exacerbations and develop targeted therapeutic interventions.

### 4.1. Potential Biomarkers of Eczema Severity

Adiponectin, a transcript of the *apM1* gene, is produced by adipose tissue. The protein is comprised of 244 amino acids. It has anti-inflammatory properties and participates in a spate of biological functions, including sensitivity to insulin and fatty acid oxidation. It also reduces the production of TNF-α, IL-6, and interferon-gamma (IFN-γ), as well as the expression of monocyte cell adhesion molecules [31]. Due to its vital relationship with lipid metabolism, adiponectin is one of the key regulators of sebocytes, keratinocytes, and skin fibroblasts as they express AdipoR1 and AdipoR2 receptors that facilitate signal transduction from the plasma membrane to the cytosolic molecular targets using the multifunctional endosomal adaptor protein phosphotyrosine, which interacts with the pleckstrin homology (PH) domain, leucine zipper-1, and AMP-activated protein kinase pathways. Adipokine upholds normal sebocyte proliferation and proper sebum composition (the factors essential for proper skin function) [32]. A decreased blood content of adiponectin has been found in psoriasis and is associated with a more frequent occurrence of metabolic syndrome in this disorder [33]. Low levels of adiponectin, akin to the present findings in adults, are present in AD children [24]. Han et al. [25] found no relationship between adiponectin and SCORAD score in a group of 64 patients aged 1–46 years, but noticed an apparent trend toward low adiponectin content in severe eczema. This trend was present in patients with extrinsic AD, as in the present study. Thus, adiponectin signaling seems to be a promising therapeutic target in AD.

Resistin is an adipose tissue-specific secretory factor named after its ability to unravel the protein’s insulin resistance enhancing property. This cysteine-rich peptide is encoded by the *RETN* gene. Resistin is mainly produced by macrophages and peripheral monocytes that are upregulated during their differentiation to macrophages [34,35]. Resistin activates pro-inflammatory cytokines through the nuclear factor-kappa beta (NF-κB) signaling pathway, including TNF-α, IL-1β, IL-6, and IL-12 [36]; the effects bear no relation to the amount of adipose tissue [37]. It also increases the expression of intercellular and vascular cell adhesion molecules [38]. Referring to the pathogenicity of AD, resistin’s ability to silence a specific immune response by the inhibition of dendritic cell function, including the uptake of native antigens and the promotion of Treg-dependent responses, is of notable significance [39]. The action of resistin is different in acute and chronic inflammation. In acute conditions, resistin enhances the macrophage-dependent response, activating the receptors for adenylyl cyclase-associated protein-1 through the mechanisms described above. In chronic inflammation, on the other hand, the release of resistin is downregulated due to the accumulation of its molecules in the endoplasmic reticulum where it acts as a chaperone, protecting cells from the response to inappropriate protein folding [39]. Machura et al. [40] have investigated the blood content of resistin in a group of 27 AD children and found, in contrast to the present findings, its increase exclusively in boys when compared to control healthy children. The authors hypothesize that the results may stem from a different sex-dependent distribution of fat tissue. Other studies on resistin in AD show, akin to the present findings, that resistin decreases in AD. Banihani et al. [41] found a significant reduction in resistin in a group of 75 AD patients (children and adults) when compared to the control group. Likewise, Farag et al. [42] have found a decrease in resistin in a group of 45 Egyptian patients (adults and children; obese patients excluded), with an inverse correlation between blood resistin content and SCORAD score, which was also found in the present study. Based on the available data, it is possible to hypothesize that consistently lower levels of resistin coexist with increased severity of AD symptoms in adults. Thus, resistin, like adiponectin, seems to be the most promising candidate to become a novel biomarker of AD severity in non-obese adults.

Leptin, a 16 kDa protein composed of 167 amino acids, is a product of the *LEP* gene and is associated with obesity. The protein is secreted by adipocytes, and aside from influencing lipogenesis, lipolysis, and angiogenesis, it also has a proinflammatory effect [43]. The main target of this polypeptide are receptors located in the ventromedial nucleus of the hypothalamus, whose activation reduces satiety and raises energy consumption. Leptin is also synthesized by keratinocytes and endothelial cells and its receptors are present in numerous peripheral tissues [44,45]. Leptin was originally thought to polarize Th lymphocytes into Th1. The immunogenic effects of leptin include the formation of proinflammatory cytokines such as TNF-α, IL-1, IL-6, and IL-8 [46], and the promotion of monocyte diapedesis through the endothelial wall involving the activation of adhesion molecules [44]. Recent studies on human leukocytes show that leptin exerts no effect on neutrophil chemotaxis; instead, leptin is a potent activator of peripheral B lymphocytes and IL-4 and IL-13 secretion by eosinophils and basophils [46]. Additionally, leptin activates dendritic cells and augments the population of CD45RA+ T-cells [47]. Aside from cytokine production, leptin induces the unfolded protein X-box binding protein-1 in the endoplasmic reticulum via the mammalian target of rapamycin and mitogen-activated protein kinase pathways, which promotes pro-allergic Th2 cell survival, which can lead to exacerbations of allergic airway disorders [48]. The immunological effects of leptin are opposite to those of resistin. The role of leptin in AD is confirmed by studies in which the rs2167270 polymorphism of the leptin gene was shown to be a risk for both increased blood leptin and AD development [22]. In accordance with that, Kimata et al. [23] demonstrated significantly elevated blood leptin in children with extrinsic AD. In the present study, we found an increased blood level of leptin in the adult AD of extrinsic origin, which lends credence to it mirroring the severity of eczema. However, there are no studies on leptin in adult AD to be compared with. The leptin issue remains contentious as other researchers failed to conclusively confirm its role in shaping atopic eczema [24,49,50] in children and there is a lack of studies in adults.

### 4.2. Other Adipokines and Body Mass

Lipocalin-2, a neutrophil gelatinase-associated lipocalin, is a 25 kDa protein made up of 178 amino acids. It shares a tertiary structure with a central hydrophobic core to bind and transport lipophilic substances, e.g., retinol, siderophores, or pheromones. It is produced in adipose tissue, with NF-κB involved in the initiation of transcription [51], but is also secreted in small amounts by the kidneys, trachea, lungs, stomach, and large intestine. The blood level of lipocalin-2 associates with adiposity and hypertriglyceridemia. It is a potent inducer of neutrophil chemotaxis and migration during inflammation and endothelial damage [52]. Additionally, lipocalin-2 has bacteriostatic activity, as it interferes with bacterial iron metabolism [51]. Various interactions of lipocalin-2 with inflammatory pathways have been demonstrated. It correlates with IL-6 and TNF-α in the blood of patients with psoriasis [53]. Interestingly, Shiratori-Hayashi et al. [54] reported the probable role of lipocalin-2 in the itch sensation in a mouse model of AD, finding that the protein activates the signal transducer and activator of transcription 3-dependent differentiation of astroglia in the posterior horns of the spinal cord, which is responsible for the enhancement of itch. In the studies by Aizawa et al. [55], it was found that blood lipocalin-2 increases in AD and psoriasis patients. Another study reports, however, a decrease in lipocalin-2 in AD [53]. The authors of both studies failed to control for body mass, which may have a bearing on the level of lipocalin-2. Irrespective of that, our present findings are somehow in line with the latter study as we noticed a decreasing trend in lipocalin-2 in adult AD patients, the majority of who were normal weight. We also found a correlation between the blood level of resistin and lipocalin-2, which may stem from the regulation of lipocalin-2 expression, partly through the NF-κB pathway, which is stimulated by resistin [39,51].

Vaspin, an adipokine produced predominantly in visceral adipose tissue, belongs to the family of serpins, i.e., serine protease inhibitors. This adipokine increases insulin sensitivity and improves glucose metabolism, particularly in the early stage of diabetes [55]. Vaspin also exerts a potent anti-inflammatory effect as it counteracts pro-inflammatory cytokine production via the inhibition of the NF-κB pathway [56,57]. It may also suppress the production of pro-inflammatory leptin and resistin [58]. A recent study has shown that vaspin is involved in the modulation of vascular endothelial function, but its role is unclear [59]. The expression of vaspin in keratinocytes is modulated by the skin immunological system [60]. The protein is involved in epidermis homeostasis as a protease inhibitor of kallikrein 7, which is crucial for the exfoliation of the epidermis [61]. The implication of vaspin in the pathogenesis of AD has not been systematically studied. In patients with psoriasis, the level of epidermal vaspin was changeable, with inappreciable changes in the blood [62]. Our present findings confirm the lack of regular changes in blood vaspin in adult AD patients, which could undermine the usefulness of its assessment in predicting the severity of eczema.

Obesity is conducive to the development of asthma with a difficult-to-control clinical phenotype [63]. The association of body mass and AD has not been well clarified. The potential association might be of high clinical importance as obesity would be a modifiable risk factor for AD. In a meta-analysis by Ali et al. [64], abdominal obesity, an element of metabolic syndrome, correlated with the occurrence of AD, particularly in women. In a study by Silverberg et al. [65] involving a group of almost 35,000 AD patients aged 18–85 from the US, a significantly higher BMI has been found when compared to the healthy population, with a distinct predilection to grade II and III obesity. In some other studies, however, the relationship between obesity and AD has not been confirmed [66,67]. Our present findings are in line with the latter studies as just three out of the 49 AD patients were overweight, and thus the pro-inflammatory propensity of the investigated adipokines, if taking part in shaping the intensity of eczema, were likely related to something other than body mass. The action of adipokines could be intertwined with other molecular mechanisms that include skin barrier dysfunction or immune abnormalities (Th2-mediated skin inflammation) [1,11,68].

## 5. Conclusions

The adult extrinsic form of atopic eczema is a disease of complex, multi-pronged, intertwined pathogenic mechanisms. While this clinical study could not adequately address the exact mechanisms of adipokines in the expression of eczema severity, we believe that we have shown that there is biological plausibility that adipokines are involved in shaping the intensity of eczema in the adult form of extrinsic atopy. Aside from the most hitherto time-tested and studied connection between the increase in blood leptin and AD symptoms, the present findings conversely show decreases in adiponectin and resistin as the presage of eczema severity. The role of these adipokines as potential biomarkers of AD requires further studies involving an array of molecular approaches. Prospective controlled trials are needed to confirm whether fluctuations in adipokine content precede or just accompany the aggravation of AD. Larger cohorts of patients than the one investigated in the present study and longitudinal follow-up study designs are required to conclusively define and validate the role of different adipokines in clinical phenotypes of such a multifaceted disease as AD. The unraveling of specific biomarkers is a sine qua non for success in the future defining of molecular AD endophenotypes and designing optimized and effective targeted treatment [69].

## Figures and Tables

**Figure 1 jcm-09-02858-f001:**
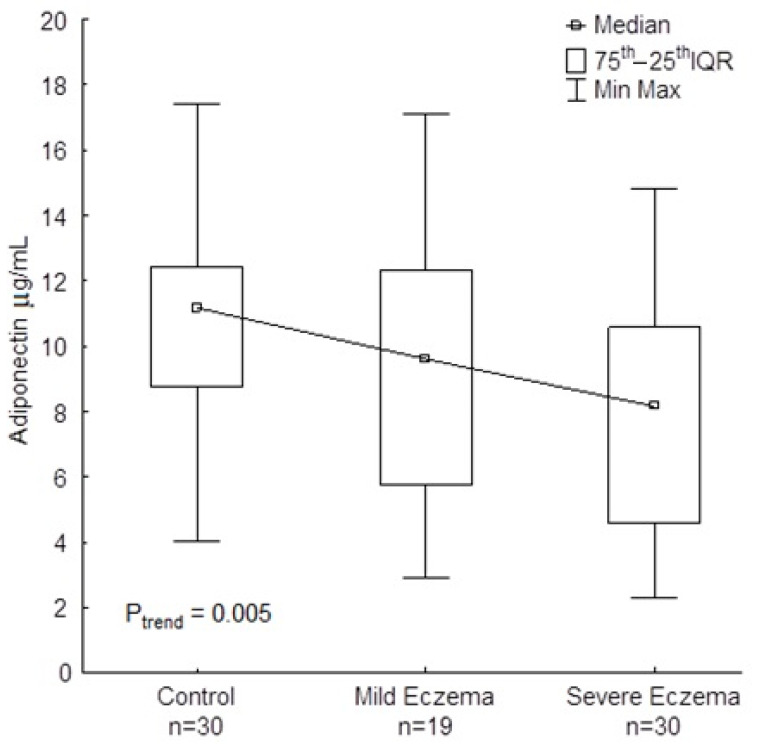
Blood adiponectin in patients mild and severe atopic eczema and healthy controls. IQR, interquartile range.

**Figure 2 jcm-09-02858-f002:**
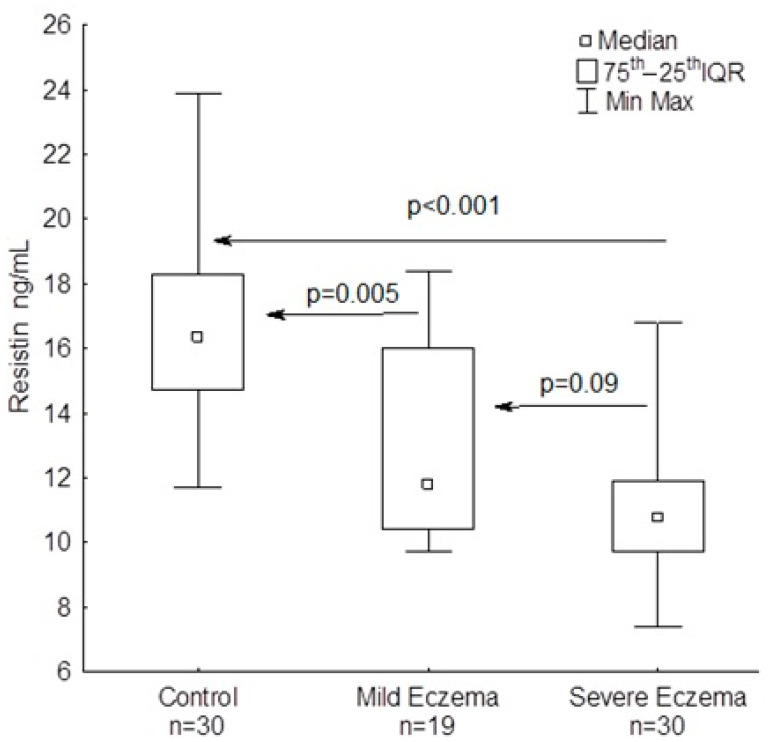
Blood resistin in patients with mild and severe atopic eczema and healthy controls.

**Table 1 jcm-09-02858-t001:** Characteristics of mild and severe atopic dermatitis patients.

	Atopic Dermatitis		Healthy Controls
	Mild Eczema (*n* = 19)	Severe Eczema (*n* = 30)	No Eczema (*n* = 30)
Sex: M/F, *n* (%)	3 (15.7)/16 (84.3)	14 (46.6)/16 (53.4)	15 (50)/15 (50)
Age: median (min–max) (years)	28 (22–60)	41 (20–75)	42 (22–78)
SCORAD: median (min–max) (points)	16.7 (10.6–20.7)	62.2 (50.2–80.4)	0.0
VAS: median (min–max) (points)	5 (3–8)	9 (8–10)	0.0
BMI: median (min–max) (kg/m^2^)	21.3 (18.5–23.4)	21.7 (18.3–28.6)	21.0 (18.9–28.7)
tIgE: median (min–max) (IU/mL)	403 (105–2300)	7670 (813–58,300)	0.0

M/F, male/female; SCORAD, SCORing Atopic Dermatitis index; VAS, visual analog scale; BMI, body mass index; tIgE, total serum IgE.

**Table 2 jcm-09-02858-t002:** Serum adipokines in the whole group of atopic dermatitis patients and healthy controls.

	Atopic Dermatitis (*n* = 49)	Controls (*n* = 30)	*p*
	Median (Min–Max)	Median (Min–Max)	
Adiponectin (µg/mL)	8.94 (2.3–17.1)	11.2 (4.0–17.4)	0.020
Resistin (ng/mL)	10.9 (9.7–18.4)	16.4 (11.7–23.9)	<0.001
Lipocalin-2 (ng/mL)	83.0 (55.6–147.6)	96.7 (52.6–178.3)	0.540
Leptin (ng/mL)	9.6 (1.9–76.3)	5.19 (0.9–188.8)	0.020
Vaspin (pg/mL)	224.3 (93.5–891.7)	267.1 (108.2–708.7)	0.540

**Table 3 jcm-09-02858-t003:** Associations among adipokine biomarkers in atopic dermatitis patients.

	Resistin	Lipocalin-2	Leptin	Vaspin
Adiponectin	−0.21	−0.18	−0.06	−0.02
Resistin	1.00	0.37 *	−0.05	0.02
Lipocalin-2		1.00	0.02	0.28
Leptin			1.00	−0.01
Vaspin				1.00

Spearman’s correlation coefficients; * *p* = 0.009.

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
