# Peer review of "Adipokines as Biomarkers of Atopic Dermatitis in Adults"

_jcm, 2020, doi:10.3390/jcm9092858_

Round 1

Reviewer 1 Report

Manuscript by Jaworel et al. investigates association of 5 adipokines in atopic dermatitis (AD) disease severity. Adipokines comprise a cytokine family expressed in white fat tissue with reports of a pathgenic role in immune and inflammatory disorders and not well known for AD. Significance was found for decreased adiponectin and resistin with respect to increased disease severity. Significance was also found for increased leptin for increased AD severity. As the cohort of 49 AD patients and 30 controls was small, it is not clear if the directionalities for the 3 adipokines for AD severity are true findings or an artifact of a small group as no validation study was performed in an independent group. Indeed, the authors provide an extensive discussion about each of the adipokines - what we know and reports of some studies to AD or other diseases. However, in its current state, the manuscript provides an incremental advance in our understanding of adipokine biomarkers for AD that was not substantiated without a validation group OR exploration of a biomarker relevance to components of SCORAD. Additional comments are provided below.

  1. Last paragraph in the introduction - The pathogenic role of adipokines in immune and inflammatory disorders could have been explained better. Is it higher or lower expression that is associated with disease?
  2. Methods - details for the ELISA experiments are needed. What were the normalizing controls? How many technical replicates for a given reading per patient sample? 
  3. How was extrinsic AD classified?
  4. Adiponectin significance - what was Ptrend value determined? 
  5. The rationale to examine the trends noted in Figure 3 are not clear. 
  6. Levels for each of the adipokine biomarker to components of SCORAD, such as features of a given AD lesion, could have been explored.
  7. What is the relationship of a given biomarkers to IgE status?

Author Response

Dear Editor,

We are much thankful for the in-depth reviews of our manuscript on atopic dermatitis. The criticism expressed by the reviewers helped us to improve the content of this research. Below please find our point-by-point responses to the reservations raised by the reviewers and the amendments made.

Reviewer I

Response to general comments.

We cannot provide any additional validation group of patients for this study other than the control group already included. Admittedly, the group of patients investigated was relatively small but the results of adipokine content outstandingly differentiate the control group from the AD patients, and the subgroups of mild and severe eczema within the AD, as based on the SCORAD assessment. Therefore, we rationalized the sample size was sufficient to stave off the possibility of a type II error. The cohort of 49 atopic dermatitis patients investigated in the present study is compliant with, or larger than, the number of patients in other similar studies on the matter. Such studies are by now scarce and the probable connection of the adipokine family with atopic eczema is just beginning to emerge. In the revision, we have improved the description of each adipokine’s relevance to the mild and severe component of SCORAD. There were clear differences in the content of adiponectin and resistin only, depending on the magnitude of SCORAD as presented in Fig. 1, 2.

Response to specific points raised by the Reviewer.

1/ The last two paragraphs of the introductions have been rephrased and expanded to describe the probable pathogenetic role of adipokines in immune and inflammatory disorders in more detail. This role, however, remains poorly understood. Concerning the majority of adipokines investigated in the study, the role is proinflammatory. On the other side, adiponectin is rather anti-inflammatory in such disorders, which is not ubiquitously confirmed.

2/ The methodological details are provided, along with the detection sensitivity of the ELISA assay for each adipokine investigated. Each sampled was assayed in duplicate and the average of the two was taken as a final result.

3/ The criteria for the external AD have been provided – second paragraph of the method.

4/ Statistical details of Ptrend calculation have been provided - last paragraph of the method.

5/ The trends shown in Fig. 3, concerning the mutual connection between changes in resistin and lipocalin-2 are mentioned in the description of results. These trends may stem from the partial involvement of NF-kappa B signaling pathway in the production of both adipokines. The figure is redundant and it does not advance the differentiation between mild and severe eczema. As such, we deleted that figure.

6/ As mentioned above, we have provided the detailed levels of each adipokine referring to the mild and severe levels of SCORAD. Only did the levels of resistin and adiponectin differ significantly in this context.

7/ There was no relationship between each adipokine biomarker and the total IgE level, which is mentioned in the description of the results of each adipokine. The only positive finding concerning the IgE was a significant but weak association between resistin and total IgE - provided in the description of resistin.

Reviewer 2 Report

This is a study on the expression of adipokines in adult atopic dermatitis. Furthermore, the authors  propose the use of some of them as biomarkers of atopic dermatitis.

In the introduction, the authors stated that atopic dermatitis is related to increasing risk of systemic diseases. The concept that  adult atopic patients have a higher frequency of systemic diseases (in particular cardiovascular diseases) is reported only by some authors, therefore the data should also report the opinion of authors who report different data.

The statement that "obesity increases the propensity for AD" is placed in the introduction in a peremptory way. Then, correctly in the discussion, the data is reported as inconstantly reported in the literature. This second option should be kept.

The final aim of the study is to propose some adipokines as biomarkers of adult atopic dermatitis. However, the data obtained are too low a number of patients, even if the author compare their results with compliant studies of the literature which however are on a limited number of patients. On the other hand, the authors write of some molecules only as promising candidates as biomarkers of atopic dermatitis, but I believe that this aspect should be further highlighted.

Finally, the need of biomarkers for diagnosis and follow-up of atopic dermatitis, especially in adults, should be emphasized (i.e. doi: 10.1016/j.jaci.2017.01.008; doi: 10.1097/ACI.0000000000000376)

Author Response

Reviewer II

Response to specific points raised by the Reviewer:

1/ The concept that atopic dermatitis is related to some other atopic and non-atopic disorders has been tackled more extensively in the first paragraph of the introduction, mentioning the pros and cons, as well as underscoring the bidirectional nature of this kind of associations.

2/ Likewise, we attempted to reduce the inadvertent peremptory connotation of the issue of obesity as a factor accompanying the AD. Obesity has indeed inconsistent and multifactorial link to AD. We have also remarked that if obesity accompanies AD, then a more frequent observation of cardiovascular disorders in AD, might be indirect due to a link between obesity and cardiovascular disorders rather than cardiovascular reason per se. In fact, in our study, we purposely focused on adult normal-weight patients suffering from AD.

3/ In the concluding paragraph of the manuscript we tackled the issues limiting the findings of this study, underscoring a relatively small sample size, a lack of longitudinal observation of changes in adipokines content, particularly of a longer follow-up period, etc. We also attempted to show more distinctly the value of establishing novel molecular biomarkers in optimizing the diagnostic process and tailoring the individual therapy in such multifaceted disorders as the AD in terms of clinical and molecular phenotypes.

Several minor inadequacies have also been corrected and a few new relevant references added. All the areas of amendments and additions are highlighted in red.

We hope that the revision will turn out satisfactory and wish to express gratitude for the reservations expressed in the reviews, which much helped improve the text.

Thank you very much.

Andrzej Jaworek

Round 2

Reviewer 1 Report

Authors have made a convincing argument for the significance of the manuscript and addressed the comments. A few minor comments are raised for accuracy.

  1. Figure 1- The y-axis label for adiponectin seems incorrect. Is it not ng/mL instead of micrograms/mL?
  2. It is more accurate to use "sex" to define the biological variable of male or female instead of "gender". This needs to be changed throughout the manuscript. 

Author Response

As requested, we substitute 'sex' for gender in the manuscript. We also thoroughly rechecked the measurement units for adiponectin. We gave the results in micrograms per mL. These units are in both Fig. 1 and Table 2. This is no mistake. The standard way to assess the content of adiponectin in human serum in the literature is micrograms per mL. Further, the results on adiponectin in the control group in our study correspond grossly to the range of normal levels reported in other studies, all in micrograms per mL (e.g., Nien et al. J Perinat Med. 2007;doi: 10.1515/JPM.2007.123).   We wish to express our thanks once again for all the comments and reservations expressed by the reviewers, which much helped improve the content of the manuscript.   Best regards, Andrzej Jaworek  
